# Recreational water exposures and illness outcomes at a freshwater beach in Toronto, Canada: A prospective cohort pilot study

Ian Young[ORCID]*, J. Johanna Sanchez, Binyam Negussie Desta, Cole Heasley[ORCID], Jordan Tustin

School of Occupational and Public Health, Toronto Metropolitan University, Toronto, Ontario, Canada

* iyoung@torontomu.ca

## Abstract

### Background

Swimming and other recreational water activities in surface waters are popular in Canada during the summer. However, these activities can also increase the risk of recreational water illness. While routine monitoring of beach water quality is conducted by local authorities each summer, little research is available in Canada about beach exposures and illness risks.

### Methods

We conducted a pilot of a prospective cohort study at a popular beach in Toronto, Ontario, Canada, in 2022 to determine characteristics of beachgoers, common water and sand exposures, the incidence of recreational water illness, and the feasibility for a larger, national cohort study. We enrolled beachgoers on-site and surveyed about their exposures at the beach and conducted a follow-up survey 7 days following their beach visit to ascertain acute gastrointestinal, respiratory, skin, ear, and eye illness outcomes. We descriptively tabulated and summarized the collected data.

### Results

We enrolled 649 households, consisting of 831 beachgoers. Water contact activities were reported by 56% of beachgoers, with swimming being the most common activity (44% of participants). Similarly, 56% of beachgoers reported digging in the sand or burying themselves in the sand. Children (≤14 years) and teenagers (15–19 years) were most likely to report engaging in water contact activities and swallowing water, while children were most likely to report sand contact activities and getting sand in their mouth. Boys and men were more likely than women and girls to report swallowing water (15.2% vs. 9.4%). Water and sand exposures also differed by household education level and participant ethno-racial identity. *E. coli* levels in beach water were consistently low (median = 20 CFU/100 mL, range = 10–58). The incidence of illness outcomes was very low (0.3–2.8%) among the 287 participants that completed the follow-up survey.

**Data Availability Statement:** The deidentified dataset and R script files for this study can be obtained at the following publicly available Github

**Funding:** Funding for this study has been made possible through a contribution from the Public Health Agency of Canada (principal investigators: IY and JT; grant number 2021-HQ-000017; website: https://www.canada.ca/en/public-health/services/funding-opportunities/infectious-diseases-climate-change-fund.html). The funders had no role in study design, data collection and analysis, decision to publish, or preparation of the manuscript.

**Competing interests:** The authors have declared that no competing interests exist.

## Conclusions

The identified beach exposure patterns can inform future risk assessments and communication strategies. Excellent water quality was observed at the studied beach, likely contributing to the low incidence of illnesses. A larger, national cohort study is needed in Canada to examine risks of illness at beaches at higher risk of fecal contamination.

## Background

Recreational water activities in surface waters are popular among Canadians during summer months. Such activities include swimming, wading, kayaking, canoeing, boating, and other water sports (e.g., surfing, jet skiing). A national survey in 2015 found that nearly 30% of Canadians reported engaging in recreational water activities in any water source in the prior 7-day period during the summer, with children aged 0–9 years more than four times as likely as adults to engage in these activities [1]. Further, approximately one-fifth of Canadians reported canoeing, kayaking, boating, or jet skiing at least once in the past 12 months in 2018 [2]. These activities can lead to exposure to various enteric and other pathogens that can cause recreational water illnesses (RWI) [3–5]. RWI has a significant health and economic burden on society. Although data are not available for Canada, approximately 90 million cases of RWI are estimated to occur each year in the United States (U.S.) due to exposures in surface waters, resulting in annual costs of US$2.2–3.7 billion [6]. Additionally, 140 RWI outbreaks due to exposure to natural surface waters were reported in the U.S. from 2000–2014, causing nearly 5,000 illnesses and two deaths [3], while 742 RWI outbreaks were reported from 1991–2007 in the Netherlands, resulting in >5,600 illnesses [7].

Globally, there is strong and consistent evidence that swimming and other high-contact recreational water activities (e.g., surfing) in surface waters increase the risk of acquiring acute gastrointestinal illness (AGI) and other RWI outcomes among beachgoers [6, 8–10]. For example, a systematic review and meta-analysis found that swimming was associated with a 2.19 times higher risk (95% CI: 1.82, 2.63) of AGI, and a 1.78 times higher risk of respiratory illness (95% CI: 1.38, 2.29), compared to those who go to the beach but do not swim [9]. In the U.S., swimming at beaches is estimated to cause 9.2 excess episodes of AGI, 8.6 excess episodes of diarrhea, and 4.9 excess days of missed work or activities per 1,000 beachgoers (representing 15%, 21%, and 9% of all such episodes, respectively) [10]. Further, beach sand contact activities (e.g., digging in the sand) have also been associated with increased risks of AGI among beachgoers [11, 12]. These risks are higher among children, as they tend to spend the most time in the water and sand, are prone to swallowing water when swimming, and have developing immune and digestive systems [10, 13–15].

While risks of RWI from different recreational water activities have been investigated in the U.S. and other countries via prospective studies [9, 10, 13, 16–18], no such studies have been conducted in Canada since 1980 [19]. In Canada, national guidelines for recreational water quality in natural water bodies are set by Health Canada, which are then adopted by local and provincial health and/or environmental authorities who conduct routine beach water surveillance [20, 21]. The guidelines set beach action values for *E. coli* and enterococci as indicators of fecal contamination and RWI risks [20]. These beach action values are based on U.S. data, given the lack of burden of illness data from Canada [20]. However, RWI risks can be affected by diverse and varying local pollution sources, environmental and weather patterns, and beachgoer activities [9, 22–24]. Therefore, Canadian data are needed on recreational water activities and associated risks of RWI. We conducted a prospective pilot cohort study of

beachgoers at a popular urban, freshwater beach in Toronto, Ontario. The study objectives were to determine characteristics of beachgoers, common water and sand exposures, the incidence of RWI, and the feasibility for a larger, national cohort study.

## Materials and methods

### Study design and setting

We conducted this study at Woodbine Beach in Toronto, Ontario, from June 4 to August 31, 2022. The study design and data collection tools were guided by the U.S. National Epidemiological and Environmental Assessment of Recreational water (NEEAR) study and the Chicago Health, Environmental Exposure, and Recreation Study (CHEERS) [13, 16–18]. We reported this manuscript in accordance with the Strengthening the Reporting of Observational Studies in Epidemiology (STROBE) guidelines (see checklist in S1 File) [25]. The Toronto Metropolitan University Research Ethics Board (REB# 2022–046) reviewed and approved this study. All participants provided informed consent via completion of a digital consent form, with parents or guardians providing informed consent for those younger than 16 years of age in their household. In addition, a digital assent form was completed for all children and youth in consultation with their parents or guardians.

Toronto is Canada's most populous city with approximately 3 million inhabitants and 6.3 million people living in the broader Toronto region. Woodbine beach is one of 11 public beaches in the city [26], located on Lake Ontario, east of Toronto's downtown core and ~1 km from the city's Ashbridges Bay Wastewater Treatment Plant [27]. We selected Woodbine beach for this pilot study because it is one of the most popular beaches in the city, with a large beachgoer population sufficient to assess cohort study recruitment feasibility. It received the Blue Flag designation in 2022 as well as prior years, an international certification program that assesses water quality and other environmental and safety criteria [28]. However, elevated *E. coli* levels occasionally occur throughout the summer season leading to swimming advisories [28, 29], suggesting one or more recurring sources of fecal contamination.

### Participation eligibility and enrollment

Our study population was all beachgoers that visited Woodbine Beach during the summer of 2022. To be eligible for participation, at least one household member aged 16 years or older needed to be present to conduct the survey in English, the household home address was required to be in Canada or the U.S., and individuals must not have already participated in the prior 21-day period. Re-enrolment was allowed after 21 days given the acute and self-limiting nature of RWI. Two data collectors attended the beach each enrollment day and approached as many beachgoers as possible. One of our aims was to determine how many beachgoers could be feasibly recruited each day to support power analysis calculations for a larger study. Household members were enrolled and surveyed together, with separate answers provided for each individual participant. We approached and enrolled potentially eligible beachgoers 2–4 days per week during the study timeframe (38 days total). Enrollment was conducted between 11AM and 5PM each day. We enrolled most Fridays, at least one weekend day, and one to two additional days per week.

### Data collection surveys

We originally planned to conduct two on-site interviews with beachgoers; an entrance survey to ask about eligibility, sociodemographic characteristics, and baseline health status, and an exit survey as participants were leaving the beach to ascertain their exposures (e.g., water and

sand contact activities). However, piloting on the first two data collection days indicated a very low proportion of households were returning to complete the exit survey (<10%). Therefore, we subsequently modified the on-site beach surveys to be completed at one point in time. Trained data collectors asked participants if they had recently arrived or if their exposure status planned to change, and if so, returned to survey those households at a later time on the same day. We conducted a follow-up survey by email or telephone (participant preference) 7 days after the beach visit to determine any RWI outcomes experienced by participants. A random draw prize ($50 gift cards) was offered as an incentive to 1% of participating households.

We adapted survey questionnaires from those used in the prior U.S. NEEAR study [18]. Additionally, we pre-tested the adapted versions with 10 individuals using a cognitive interviewing approach [30]. We used this process to refine the questions, their clarity, and layout. Participants completed on-site beach surveys via our tablet devices. The questionnaires were hosted on the web-based SimpleSurvey platform (OutSideSoft Solutions Inc., Quebec, Canada). We sent up to five reminders (emails or phone calls) to participants that did not complete the follow-up survey for 3–5 days following the initial contact. Copies of the questionnaires are available in S2 File.

## Exposures and outcomes of interest

We collected data on two different beach exposures of interest: water contact activities and sand contact activities. Specifically, we asked participants to indicate if they swam, waded below their waist, or engaged in various other recreational activities (e.g., paddle-boarding, kiteboarding, boating). We also asked participants whether they submerged their face in the water and if they swallowed any water. For sand activities, we asked participants to indicate if they dug or buried themselves in the sand or got sand in their mouth.

We collected household-level information on province or state of residence, income, and the highest educational attainment level within the household. We then collected the following additional sociodemographic information for each individual participant: age group, sex at birth, current gender, and ethno-racial identity. We also collected self-reported information about baseline health outcomes, chronic health condition status, and other beach exposures (e.g., food consumption, sunscreen application). Sociodemographic questions were optional, so resulted in different rates of missing values for those questions, while all other questions on the survey were mandatory and completed by all participants.

The follow-up survey asked participants indicate if they or any other household members experienced any of the following RWI outcomes of interest within 7 days of their beach visit: AGI, respiratory, skin, ear, and eye infections. We selected this length of follow-up as it corresponds with incubation periods of the primary viral and bacterial enteric pathogens of concern for RWI [31, 32]. We defined AGI as: (a) diarrhea (≥3 loose stools in 24 hrs); (b) vomiting; (c) nausea with stomach cramps; or (d) nausea or stomach cramps that interfere with regular daily activities (e.g., missed work or school) [10, 13, 16, 17, 33]. We defined other RWI outcomes as follows: respiratory illness (fever with sore throat, fever with nasal congestion, or cough with phlegm); skin infection (rash or itchy skin); ear infection or earache; and eye infection or irritation [13, 16, 17, 33, 34].

We obtained data on beach water *E. coli* levels during our enrollment dates from the local public health authority (City of Toronto Open Data Portal). The City of Toronto conducts routine sampling for *E. coli* once per day during the summer season at all beaches, including Woodbine [21, 26]. Water samples are collected in sterile bottles at a depth of 1–1.5 m, approximately 15–30 cm below the surface, at five points along the beach [21]. Samples are shipped to the provincial laboratory network within 24 hr and are tested using a standard, culture-based

membrane filtration method [35]. The geometric mean of the five samples is used as a indicator of water quality, with the city using a locally adopted threshold of ≤100 CFU/100 mL to determine whether to issue a swimming advisory [26].

## Data analysis

We downloaded data from SimpleSurvey and imported it into RStudio (version 2022.07.1 running R 4.2.1) for formatting, preparation, and analysis [36, 37]. Given the low incidence of illness outcomes reported in this study and low completion rate for the follow-up survey, we focused on descriptively summarizing the survey results instead of modelling exposure-illness associations. We calculated descriptive tabulations and summary measures for all exposure, sociodemographic, and outcome variables. Participants who had any of the RWI outcomes at baseline were excluded from our summary of those outcomes. We conducted cross-tabulations to compare water and sand exposures by age group, gender identity, highest education level in the household, and ethno-racial identity. Further, we compared sociodemographic characteristics of those who completed and did not complete the follow-up survey to evaluate possible biases from loss to follow-up. The deidentified dataset and R script files are available at the following Github page: https://github.com/iany33/beachcohortpilot2022.

## Results

### Participant characteristics

Throughout the enrollment period we approached 1,007 households, of which 649 (64.4%) agreed to participate. Among the 348 (35.6%) who did not participate, primary reasons for not participating included: not interested in the study (N = 151, 42.2%), did not want to be bothered or interrupted at the beach (N = 128, 35.8%), did not speak English (N = 25, 7.0%), did not have time (N = 22, 6.1%), not eligible (N = 18, 5.1%), and wanted to remain anonymous (N = 14, 3.9%). Of participating households, 831 individual participants completed the beach survey and 287 completed the follow-up survey, for a follow-up rate of 34.5%. Most participants requested the email follow-up option (78.7%), which had a slightly higher completion rate than telephone follow-ups (35.3% vs. 32.1%). Only two participants completed the survey more than once. The average time of completion of the beach and follow-up surveys was 4 min 56 sec and 1 min 57 sec, respectively.

The characteristics of participants that completed the beach survey, stratified by enrollment month, are shown in Table 1. Most participants were aged 20–39 (56.7%) and identified as woman or girls (65.2%). Most participants (60.2%) chose not to answer the question about household income. Nearly all participants (93.0%) resided in Ontario, and most participants (57.4%) identified as white. The most prevalent baseline illness was respiratory illness (1.9%). Over 40% of participants reported engaging in other recreational water contact within the prior 2-week period. Fewer children, households in the lowest income categories, and households with post-graduate degrees were enrolled in July and August compared to June. Fewer illness outcomes were also identified in July and August compared to June.

### Beach exposures

A summary of the water and sand contact exposures and activities reported by participants is shown in Table 2, stratified by enrollment month. Swimming and wading were both lowest in June, with swimming increasing in July (46.0%) and highest in August (61.0%). The number of participants reporting wading was similar in July and August (31–32%). Similarly, the prevalence of face contact with water and swallowing water also increased monthly, with highest

**Table 1. Characteristics of 831 beachgoers surveyed at Woodbine Beach, Toronto, 2022, stratified by enrollment month.** AGI = acute gastrointestinal illness.

| Characteristic | N (%) | | | |
| --- | --- | --- | --- | --- |
| | June | July | August | Total |
| Age group: | | | | |
| 0–14 | 55 (20.5) | 25 (9.5) | 23 (7.9) | 103 (12.5) |
| 15–19 | 31 (11.6) | 50 (18.9) | 57 (19.7) | 138 (16.8) |
| 20–39 | 142 (53.0) | 155 (58.7) | 169 (58.3) | 466 (56.7) |
| 40+ | 40 (14.9) | 34 (12.9) | 41 (14.1) | 115 (14.0) |
| Missing | 3 | 1 | 5 | 9 |
| Gender identity: | | | | |
| Woman/girl | 174 (64.7) | 154 (59.7) | 205 (70.4) | 533 (65.2) |
| Man/boy | 89 (33.1) | 99 (38.4) | 82 (28.2) | 270 (33.0) |
| Gender fluid, non-binary, or transgender | 6 (2.2) | 5 (1.9) | 4 (1.4) | 15 (1.8) |
| Missing | 2 | 7 | 4 | 13 |
| Household income (CAD$): | | | | |
| <$40,000 | 44 (21.2) | 17 (16.8) | 1 (4.5) | 62 (18.7) |
| $40,000–79,999 | 39 (18.8) | 37 (36.6) | 6 (27.3) | 82 (24.8) |
| $80,000–149,999 | 72 (34.6) | 32 (31.7) | 12 (54.5) | 116 (35.0) |
| $150,000+ | 53 (25.5) | 15 (14.9) | 3 (13.6) | 71 (21.5) |
| Missing | 63 | 164 | 273 | 500 |
| Highest education completed in household: | | | | |
| High school or less | 58 (22.6) | 58 (36.0) | 70 (41.2) | 186 (31.6) |
| College, trades, or apprenticeship | 49 (19.1) | 27 (16.8) | 43 (25.3) | 119 (20.2) |
| Bachelor's degree | 64 (24.9) | 32 (19.9) | 57 (33.5) | 153 (26.0) |
| Post-graduate degree | 86 (33.5) | 44 (27.3) | 0 (0.0) | 130 (22.1) |
| Missing | 14 | 104 | 125 | 243 |
| Location of residence: | | | | |
| Ontario | 249 (91.9) | 241 (90.9) | 283 (95.9) | 773 (93.0) |
| U.S. | 14 (5.2) | 12 (4.5) | 7 (2.4) | 33 (4.0) |
| Quebec | 5 (1.8) | 9 (3.4) | 2 (0.7) | 16 (1.9) |
| Other Canadian provinces[a] | 3 (1.2) | 3 (1.1) | 3 (1.0) | 9 (1.1) |
| Ethno-racial identity: | | | | |
| White | 130 (52.2) | 135 (60.5) | 146 (59.8) | 411 (57.4) |
| South Asian | 18 (7.2) | 22 (9.9) | 21 (8.6) | 61 (8.5) |
| Southeast Asian | 9 (3.6) | 15 (6.7) | 24 (9.8) | 48 (6.7) |
| Arab or Middle Eastern | 15 (6.0) | 14 (6.3) | 18 (7.4) | 47 (6.6) |
| East Asian | 19 (7.6) | 11 (4.9) | 11 (4.5) | 41 (5.7) |
| Latin | 20 (8.0) | 9 (4.0) | 5 (2.0) | 34 (4.7) |
| Black | 13 (5.2) | 7 (3.1) | 13 (5.3) | 33 (4.6) |
| Multiple ethnicities | 16 (6.4) | 9 (4.0) | 6 (2.5) | 31 (4.3) |
| Indigenous | 9 (3.6) | 1 (0.4) | 0 (0.0) | 10 (1.4) |
| Missing | 22 | 42 | 51 | 115 |
| Baseline illness status[b]: | | | | |
| Respiratory illness | 8 (3.0) | 7 (2.6) | 1 (0.3) | 16 (1.9) |
| AGI | 5 (1.8) | 1 (0.4) | 1 (0.3) | 7 (0.8) |
| Skin infection | 4 (1.5) | 1 (0.4) | 1 (0.3) | 6 (0.7) |
| Ear infection | 1 (0.4) | 0 (0.0) | 0 (0.0) | 1 (0.1) |
| Eye infection | 0 (0.0) | 1 (0.4) | 0 (0.0) | 1 (0.1) |
| Baseline health conditions[b]: | | | | |

*(Continued)*

**Table 1.** (Continued)

| Characteristic | N (%) | | | |
|---|---|---|---|---|
| | June | July | August | Total |
| Allergies | 16 (5.9) | 25 (9.4) | 30 (10.2) | 71 (8.5) |
| Chronic respiratory condition | 7 (2.6) | 8 (3.0) | 12 (4.1) | 27 (3.2) |
| Chronic gastrointestinal condition | 8 (3.0) | 6 (2.3) | 1 (0.3) | 15 (1.8) |
| Immune-compromised | 3 (1.1) | 5 (1.9) | 2 (0.7) | 10 (1.2) |
| Engaged in other recreational water activities within the past 2 weeks | 85 (31.4) | 122 (46.0) | 141 (47.8) | 348 (41.9) |

[a] Other provinces included Alberta (N = 5), Nova Scotia (N = 2), and British Columbia and Newfoundland and Labrador (N = 1 each).

[b] Multiple selections were possible for these variables.

levels in August (Table 2). In contrast, digging in the sand (58.7%) and getting sand in one's mouth (29.2%) were most commonly reported in June, while burying oneself in the sand was most common in July (29.1%). In addition to these exposures, 37.5% (N = 312) of participants reported having contact with seaweed, 73.8% (N = 613) reported applying sunscreen, and 69.4% (N = 577) reported consuming food at the beach.

Tables 3–6 show differences in the most common beach exposures by participant sociodemographic characteristics. Children (0–14) and teenagers (15–19) were most likely to have any water contact, while teenagers swam more than other age groups (Table 3). Children and teenagers were also most likely to have face contact with water and swallow water. Almost all children reported sand contact (91.3%); they were the most likely age group to engage in sand contact activities and to get sand in their mouth (Table 3).

Boys and men engaged in similar water contact activities as woman and girls, but they were more likely to have their faces touch the water and report swallowing water (Table 4). Boys and men were also more likely to report digging in the sand, but there were no differences in the likelihood of getting sand in one's mouth. Participants from households where the highest

**Table 2. Water and sand exposures reported among beachgoers, stratified by study enrollment month, at Woodbine Beach, Toronto, 2022.**

| Exposure | N (%) | | | |
|---|---|---|---|---|
| | June | July | August | Total |
| Water contact | 106 (39.1) | 152 (57.4) | 209 (70.8) | 467 (56.2) |
| Swimming | 65 (24.0) | 122 (46.0) | 180 (61.0) | 367 (44.2) |
| Wading (below one's waist) | 43 (15.9) | 85 (32.1) | 92 (31.2) | 220 (26.5) |
| Paddleboarding | 6 (2.2) | 5 (1.9) | 2 (0.7) | 13 (1.6) |
| Other water sports[a] | 4 (1.5) | 1 (0.4) | 2 (0.7) | 7 (0.8) |
| Other minimal contact activities[b] | 6 (2.2) | 3 (1.1) | 2 (0.7) | 11 (1.3) |
| Face contact with water | 28 (10.3) | 68 (25.7) | 90 (30.5) | 186 (22.4) |
| Swallowing water | 15 (5.5) | 29 (10.9) | 50 (16.9) | 94 (11.3) |
| Sand contact[c] | 168 (62.0) | 171 (64.5) | 127 (43.1) | 466 (56.1) |
| Digging in the sand | 159 (58.7) | 125 (47.2) | 122 (41.4) | 406 (48.9) |
| Burying oneself in the sand | 45 (16.6) | 77 (29.1) | 13 (4.4) | 135 (16.2) |
| Sand in mouth | 52 (29.2) | 30 (16.8) | 25 (19.7) | 107 (22.1) |

[a] Other water sports included kitesurfing (N = 3), diving (N = 2), surfing (N = 1), and waterskiing (N = 1).

[b] Other minimal contact activities included kayaking (N = 4), boating (N = 3), fishing (N = 2), sailing and canoeing (N = 1 each).

[c] We defined sand contact as digging in the sand or burying oneself in the sand.

**Table 3. Water and sand exposures reported among beachgoers, stratified by age group, Woodbine Beach, Toronto, 2022.**

| Exposure | N (%) | | | |
|---|---|---|---|---|
| | 0–14 years (N = 103) | 15–19 years (N = 138) | 20–39 years (N = 466) | 40+ years (N = 115) |
| Water contact | 71 (68.9) | 89 (64.5) | 236 (50.6) | 67 (58.3) |
| Swimming | 46 (44.7) | 84 (60.9) | 182 (39.1) | 52 (45.2) |
| Wading (below one's waist) | 38 (36.9) | 29 (21.0) | 112 (24.0) | 38 (33.0) |
| Face contact with water | 33 (32.0) | 48 (34.8) | 79 (17.0) | 25 (21.7) |
| Swallowing water | 17 (16.5) | 21 (15.2) | 46 (9.9) | 9 (7.8) |
| Sand contact[a] | 94 (91.3) | 79 (57.2) | 230 (49.4) | 60 (52.2) |
| Digging in the sand | 90 (87.4) | 67 (48.6) | 197 (42.3) | 49 (42.6) |
| Burying oneself in the sand | 34 (33.0) | 21 (15.2) | 59 (12.7) | 20 (17.4) |
| Sand in mouth | 45 (47.9) | 14 (17.5) | 36 (14.6) | 11 (18.0) |

[a] We defined sand contact as digging in the sand or burying oneself in the sand.

**Table 4. Water and sand exposures reported among beachgoers, stratified by gender identity, Woodbine Beach, Toronto, 2022.**

| Exposure | N (%) | | |
|---|---|---|---|
| | Woman/girl (N = 533) | Man/boy (N = 270) | Fluid/non-binary/trans (N = 15) |
| Water contact | 290 (54.4) | 159 (58.9) | 8 (53.3) |
| Swimming | 223 (41.8) | 127 (47.0) | 8 (53.3) |
| Wading (below one's waist) | 137 (25.7) | 77 (28.5) | 2 (13.3) |
| Face contact with water | 96 (18.0) | 78 (28.9) | 5 (33.3) |
| Swallowing water | 50 (9.4) | 41 (15.2) | 0 (0.0) |
| Sand contact[a] | 286 (53.7) | 163 (60.4) | 9 (60.0) |
| Digging in the sand | 246 (46.2) | 145 (53.7) | 8 (53.3) |
| Burying oneself in the sand | 86 (16.1) | 44 (16.3) | 1 (6.7) |
| Sand in mouth | 65 (21.9) | 33 (19.4) | 3 (33.3) |

[a] We defined sand contact as digging in the sand or burying oneself in the sand.

**Table 5. Water and sand exposures reported among beachgoers, stratified by highest education completed in the household, Woodbine Beach, Toronto, 2022.**

| Exposure | N (%) | | | |
|---|---|---|---|---|
| | High school or less (N = 186) | College/trades/ apprenticeship (N = 119) | Bachelor's degree (N = 153) | Post-graduate (N = 130) |
| Water contact | 104 (55.9) | 74 (62.2) | 83 (54.2) | 50 (38.5) |
| Swimming | 85 (45.7) | 55 (46.2) | 60 (39.2) | 33 (25.4) |
| Wading (below one's waist) | 44 (23.7) | 27 (22.7) | 29 (19.0) | 20 (15.4) |
| Face contact with water | 43 (23.1) | 30 (25.2) | 31 (20.3) | 14 (10.8) |
| Swallowing water | 21 (11.3) | 11 (9.2) | 16 (10.5) | 4 (3.1) |
| Sand contact[a] | 96 (51.6) | 62 (52.1) | 71 (46.4) | 80 (61.5) |
| Digging in the sand | 84 (45.2) | 59 (49.6) | 67 (43.8) | 76 (58.5) |
| Burying oneself in the sand | 22 (11.8) | 15 (12.6) | 11 (7.2) | 19 (14.6) |
| Sand in mouth | 23 (22.8) | 11 (17.2) | 14 (17.7) | 17 (20.5) |

[a] We defined sand contact as digging in the sand or burying oneself in the sand.

**Table 6. Water and sand exposures reported among beachgoers, stratified by ethno-racial identity, Woodbine Beach, Toronto, 2022.**

| Exposure | N (%) | | | | | | | | |
|---|---|---|---|---|---|---|---|---|---|
| | Arab (N = 47) | Black (N = 33) | East Asian (N = 41) | Indigenous (N = 10) | Latin (N = 34) | Multiple ethnicities (N = 31) | South Asian (N = 61) | Southeast Asian (N = 48) | White (N = 411) |
| Water contact | 31 (66.0) | 14 (42.4) | 17 (41.5) | 3 (30.0) | 14 (41.2) | 23 (74.2) | 33 (54.1) | 22 (45.8) | 238 (57.9) |
| Swimming | 26 (55.3) | 12 (36.4) | 11 (26.8) | 3 (30.0) | 13 (38.2) | 12 (38.7) | 23 (37.7) | 21 (43.8) | 191 (46.5) |
| Wading (below one's waist) | 13 (27.7) | 5 (15.2) | 7 (17.1) | 1 (10.0) | 3 (8.8) | 12 (38.7) | 23 (37.7) | 12 (25.0) | 120 (29.2) |
| Face contact with water | 14 (29.8) | 6 (18.2) | 6 (14.6) | 0 (0.0) | 5 (14.7) | 7 (22.6) | 12 (19.7) | 5 (10.4) | 99 (24.1) |
| Swallowing water | 6 (12.8) | 4 (12.1) | 5 (12.2) | 0 (0.0) | 4 (11.8) | 2 (6.5) | 12 (19.7) | 4 (8.3) | 49 (11.9) |
| Sand contact[a] | 26 (55.3) | 17 (51.5) | 26 (63.4) | 6 (60.0) | 22 (64.7) | 23 (74.2) | 31 (50.8) | 26 (54.2) | 227 (55.2) |
| Digging in the sand | 18 (38.3) | 16 (48.5) | 23 (56.1) | 6 (60.0) | 20 (58.8) | 20 (64.5) | 26 (42.6) | 21 (43.8) | 195 (47.4) |
| Burying oneself in the sand | 12 (25.5) | 1 (3.0) | 7 (17.1) | 2 (20.0) | 8 (23.5) | 5 (16.1) | 11 (18.0) | 7 (14.6) | 75 (18.2) |
| Sand in mouth | 5 (19.2) | 1 (5.9) | 4 (14.8) | 1 (16.7) | 7 (31.8) | 9 (39.1) | 7 (21.2) | 7 (25.0) | 58 (24.3) |

[a] We defined sand contact as digging in the sand or burying oneself in the sand.

level of education completed was a post-graduate degree were less likely than those in other households to report water contact activities and exposures, and they were also more likely to report sand contact exposures (Table 5).

Differences in exposures by ethno-racial identity are shown in Table 6. Although there are small numbers in most categories, some trends are noted. Arab and Middle Eastern participants reported the highest prevalence of swimming and face contact with water, while East Asian participants were least likely to report swimming (Table 6). Participants that identified as multiple ethnicities, Latin, and East Asian were most likely to report engaging in sand contact activities (Table 6).

Across the study enrollment period, the beach water geometric mean *E. coli* levels were consistently low and never exceeded the local health threshold level (median = 20, IQR = 21, range = 10–58). Fig 1 shows the distribution of *E. coli* level exposures by participant.

## Illness outcomes

Table 7 shows the incidence of each of the five illness outcomes of interest among those who completed the follow-up survey, stratified by water and sand contact exposure status. The most common outcome was skin infection (2.8%), which was more frequently reported among those who had sand contact exposures. Similarly, all five cases of respiratory illness reported sand contact exposures. Only three cases of AGI were identified, and one each of ear and eye infections. Severity outcomes were collected for the AGI cases only: one reported missing work due to illness, all three reported taking over-the-counter drugs to treat illness, and none reported seeking medical attention.

A comparison of sociodemographic and baseline characteristics of those who completed the follow-up survey vs. those who did not are shown in S1 Table. Notably, fewer children and more older adults, those with higher incomes, and those with a bachelor's vs. high school or less education were more likely to complete the follow-up survey (S1 Table). Those who completed the follow-up survey also were more likely to report water contact (62.7% vs. 52.8%), and less likely to report burying themselves in the sand (10.8% vs. 19.1%) and getting sand in their mouth (14.1% vs. 26.2%).

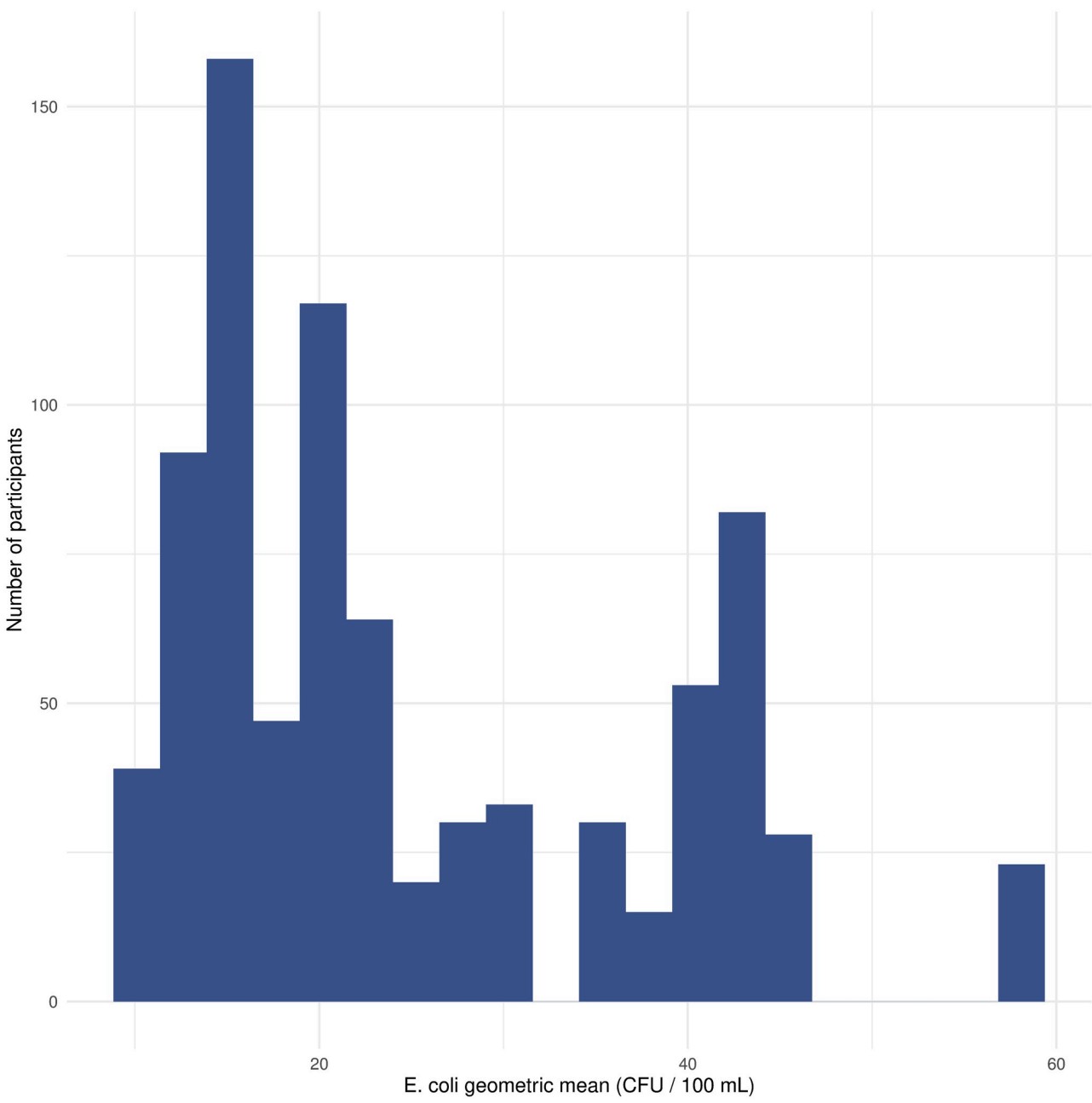

**Fig 1. Histogram of geometric mean *E. coli* level exposures (CFU / 100 mL) by participant, Woodbine Beach, Toronto, 2022.**

## Discussion

We conducted a prospective cohort pilot study at a popular, urban beach in Toronto, Canada, to determine beachgoer characteristics, reported types of water and sand exposures, the incidence of RWI, and the feasibility for a national cohort study. The findings of this study provide insights into beach exposure patterns and illness outcomes at a freshwater beach with very low levels of identified fecal pollution. In addition, we provided a detailed examination of differences in water and sand exposures by beachgoer characteristics, which can inform future

**Table 7. Incidence of RWI outcomes among beachgoers, stratified by water and sand contact status, at Woodbine Beach, Toronto, 2022[a].** AGI = acute gastrointestinal illness.

| Outcome | N (%) | | | | |
|---|---|---|---|---|---|
| | Water contact (N = 180) | No water contact (N = 107) | Sand contact (N = 155) | No sand contact (N = 132) | Total (N = 287) |
| AGI | 2 (1.1) | 1 (0.9) | 2 (1.3) | 1 (0.8) | 3 (1.0) |
| Respiratory illness | 2 (1.1) | 3 (2.8) | 5 (3.2) | 0 (0) | 5 (1.7) |
| Skin infection | 3 (1.7) | 5 (4.7) | 6 (3.9) | 2 (1.5) | 8 (2.8) |
| Ear infection | 1 (0.6) | 0 (0) | 1 (0.6) | 0 (0) | 1 (0.3) |
| Eye infection | 0 (0) | 1 (0.9) | 0 (0) | 1 (0.8) | 1 (0.3) |

[a] Incidence values exclude those who had the same outcome at baseline.

studies and risk communication efforts. Our sample was approximately two-thirds women and girls, and it is not clear whether this reflected the wider beachgoer population or if women and girls were more likely to participate in the study. Previous U.S. cohort studies also had more female than male participants [13, 18, 38, 39]. Based on the most recently available Census data (2021), our sample had more participants with a post-graduate degree than with high school or less education and contained more individuals that identified as white vs. other ethno-racial backgrounds compared to the average for Toronto [40]. Future research should investigate ways to enroll more diverse samples in beach cohort studies, particularly from historically underrepresented population groups.

We identified much lower levels of water contact and swimming, and higher levels of sand contact activities, compared to prior cohort studies in the U.S. [14, 22]. Water and sand contact activities in August more closely matched prior studies, suggesting that the colder lake water temperatures in Toronto result in different exposure patterns for beachgoers than in other locations. Earlier in the beach season in Toronto (June and July), sand exposures appear to be the most important risk of encountering possible pathogens among beachgoers, especially children. Sand exposures have been previously associated with increased risks of AGI among beachgoers [11, 12]. However, there is no routine monitoring for fecal indicator organisms in beach sand in Toronto or other jurisdictions in Ontario, nor are there any guidelines for such monitoring in Canada [20]. Instead, Health Canada recommends that authorities primarily manage possible sand contamination through control measures such as beach grooming and access restrictions [41]. In their 2021 recreational water guidelines, the World Health Organization suggested a provisional health threshold level of <60 CFU/g of intestinal enterococci for beach sand [42]. However, they also noted that additional research is needed to determine acceptable levels of fecal contamination in sand and to assess possible RWI risks from exposure [42]. In addition to enteric pathogens, both beach water and sand can also contain antimicrobial-resistant (AMR) bacteria (e.g., methicillin-resistant *Staphylococcus aureus*) and opportunistic fungi that can also present different risks of illness to beachgoers [43–45]. Neither beach water nor sand are monitored for these microbial hazards in Canada. Additional research is needed on beach sand contamination, possible risks to beachgoers from sand exposures, and contamination levels and risks from non-fecal microbial hazards in sand and water.

We found that children and teenagers were most likely to report any water contact activities and swallowing water, while nearly all children engaged in sand contact activities and nearly half reported getting sand in their mouth. These exposure trends are similar to prior U.S. studies [14, 22], though our study found a much higher incidence of sand contact activities and getting sand in one's mouth, further highlighting the need for additional investigation of the microbial quality of beach sand. We found no gender-related differences in water contact

activities but found that boys and men were more likely to report having their faces touch water, swallow water, and dig in the sand. This finding corresponds with prior research in the U.S. that found that boys tend to swallow a greater volume of water and spend more time in the water than girls [14]. Additional research to investigate gender-related differences in beach exposures is warranted to guide risk communication efforts.

We found that participants from higher education households were less likely to report water contact activities and more likely to report sand contact activities. This finding could be related to a greater understanding of or attention to beach water quality risks among beachgoers with higher levels of household education [46, 47], but further research is needed in this area. Similarly, we found some difference in beach activities and exposures depending on participants reported ethno-racial identity, which could indicate possible cultural differences in beach behaviours. Additional research into these differences is necessary to inform targeted beach water quality education and outreach needs and gaps among equity-deserving population groups.

*E. coli* levels were consistently very low during the study period, with none of our participants exposed to levels exceeding local or federal beach action values [20, 26]. It was not surprising, therefore, that we identified very few cases of AGI among respondents that completed the follow-up survey. This finding suggests that there was no increased risk of enteric illness among beachgoers at this urban beach, given the excellent water quality levels observed during the study period. However, we identified very low numbers of all RWI outcomes at baseline and in the follow-up survey, indicating more research is needed to examine possible relationships between water and sand exposures and RWI outcomes among beachgoers in Canada. It is possible that the lower RWI rates may also be due to enrollment of fewer children than in prior studies, especially during July and August, given that children are at a much higher risk of contracting AGI and other RWI outcomes [10, 48]. Some of the illness outcomes (e.g., eye infections) were observed at higher levels in the non-exposed groups, which could have been due to the low sample size, potential misclassification bias, or confounding factors. Future cohort studies should include beach sites that have a history of more fluctuations in water quality, including numerous days with high fecal contamination levels, and sites with known sources of human sewage contamination (e.g., sewage outfalls) to assess the relationship between water quality levels and RWI. Additionally, in this study we relied on culture-based *E. coli* levels as an indicator of fecal contamination, as this is the indicator routinely used by public health authorities in Ontario [35]. Future studies should aim to incorporate rapidly-measured indicators, such as qPCR-based enterococci, and microbial source tracking genetic markers that can determine specific sources of fecal contamination [49, 50].

Fewer children completed our follow-up survey than expected, which likely explains the other identified differences in reported exposure levels found between those who completed the follow-up compared to those who did not. This is likely at least in part due to parents and guardians of children being too busy to complete the follow-up survey, as well as completing the online survey for themselves only and not including or forgetting to include the names and outcomes for their children. For example, our data collectors anecdotally noted that parents and caregivers were often reluctant to take time to participate and complete surveys on the beach given their childcare duties, contributing to the responses of 'not wanting to be bothered' and 'not having time' to participate among 41.9% of non-participants. Future beach cohort studies should incorporate more comprehensive methods to enroll and boost retention of families with children. Targeted prompts and personalized follow-ups and requests for additional information from such participants might help to address such data gaps [51]. Further, as this was a pilot study with limited funding, we were only able to offer a chance at a draw prize for participating households. Providing direct monetary incentives at the beach to all

participating households in future studies would also likely enhance enrollment and follow-up retention among all participants, including families with children [52, 53].

There are several limitations of this study. While we had a relatively high participation rate among approached households in this study (64%), the follow-up survey completion rate was lower than prior cohort studies conducted in the U.S. which ranged widely from ~45–95% [16, 17, 33, 38, 39]. This limits conclusions that can be drawn from the illness outcome data. Combining the beach entrance and exit survey was also not ideal, but we determined was necessary after the first two enrollment days given that most participants were not returning to complete the exit survey. Woodbine Beach has no defined entrance and exit points, with beachgoers arriving and leaving from a boardwalk along a wide beach area, which likely contributed to this issue. Therefore, our modified strategy ensured that we collected all beach-level data from participants as they were finishing up or nearing the end of their beach visit. Although additional misclassification errors were possible, we also asked participants to report if their exposure status changed after completing the survey, and very few (~5%) reported any changes. We intended to capture a range of different water exposures, but nearly all reported activities were swimming or wading. More targeted efforts may be needed in future studies to enroll beachgoers engaging in other water sport and minimal contact activities.

## Conclusion

This pilot cohort study identified and categorized water and sand contact exposures among beachgoers at a popular urban beach in Toronto, Canada. Sand contact activities were much more common than water activities compared to prior studies, especially earlier in the bathing season (June and July). Future research is warranted to investigate the microbial quality and risks of illness from sand exposures. Differences in exposure patterns were noted by age group, gender identity, education, and ethno-racial identity, and those results can be used to guide targeted risk communication and outreach initiatives as well as quantitative microbial risk assessments of beach water quality risks. While follow-up data were also collected on RWI, a high attrition rate and low baseline levels limits the utility of these outcomes. The beach examined in this study had excellent water quality as indicated by *E. coli* monitoring data and its Blue Flag certification status, and any health risks to bathers were likely minimal. A larger, national cohort study is needed in Canada to examine relationships between water and sand contact exposures, water quality indicators, and RWI outcomes in beaches at high risk of fecal pollution.

## Supporting information

**S1 File. STROBE checklist for cohort studies.**
(DOCX)

**S2 File. Beach cohort study questionnaires, Woodbine Beach, Toronto, 2022.**
(DOCX)

**S1 Table. Comparison of participants who completed the follow-up survey vs. those who did not complete the follow-up survey, Woodbine Beach, Toronto, 2022.**
(DOCX)

## Acknowledgments

The authors thank all of the beachgoers who offered their time in completing questionnaires for this study. We also thank Tannis Bubeloff and Zacharia Ramdayal for their efforts conducting participant enrollment, administering surveys, and coordinating follow-ups.

## Author Contributions

**Conceptualization:** Ian Young, J. Johanna Sanchez, Jordan Tustin.

**Data curation:** Ian Young, Jordan Tustin.

**Formal analysis:** Ian Young.

**Funding acquisition:** Ian Young, J. Johanna Sanchez, Jordan Tustin.

**Investigation:** Ian Young, J. Johanna Sanchez, Binyam Negussie Desta, Jordan Tustin.

**Methodology:** Ian Young, J. Johanna Sanchez, Binyam Negussie Desta, Jordan Tustin.

**Project administration:** Ian Young, J. Johanna Sanchez, Binyam Negussie Desta, Cole Heasley, Jordan Tustin.

**Resources:** J. Johanna Sanchez, Cole Heasley.

**Software:** Cole Heasley.

**Supervision:** Ian Young, Binyam Negussie Desta, Jordan Tustin.

**Writing – original draft:** Ian Young.

**Writing – review & editing:** Ian Young, J. Johanna Sanchez, Binyam Negussie Desta, Cole Heasley, Jordan Tustin.

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
