## [Decision Letter · Decision Letter 0]

10 Feb 2023

PONE-D-22-34376Recreational water exposures and illness outcomes at a freshwater beach in Toronto, Canada: a prospective pilot cohort studyPLOS ONE

Dear Dr. Young,

Thank you for submitting your manuscript to PLOS ONE. After careful consideration, we feel that it has merit but does not fully meet PLOS ONE’s publication criteria as it currently stands. Therefore, we invite you to submit a revised version of the manuscript that addresses the points raised during the review process.

We look forward to receiving your revised manuscript.

Kind regards,

Zakir Abdu

Academic Editor

PLOS ONE

Journal Requirements:

Additional Editor Comments:

Dear Author (s),

Thank you for submitting an important and scientific piece of work in the field of SRH Recreational water exposures and illness outcomes at a freshwater beach in PLOS ONE journal. With due reverence to our reviewers and valuable comments, the following points are additional ones found to be addressed, the manuscript writing and its flow are praiseworthy though :

Clarify the methodology part. Who was your study population? Source population? And what is your study unit? How did you select one (Woodbine beach) from the popular 11? (is popularity is a criteria? I did not think so). Please justify? What is the others beach?While collecting data, what do think about more than 1 participants present in household?Clearly describe the sample size calculation with rationaleAnother of my concern is that about confidentiality (while emailing or calling the participants for the issue of get information)Did you found a participant with illness? What did you do for them?

Reviewers' comments:

Reviewer's Responses to Questions

**Comments to the Author**

1. Is the manuscript technically sound, and do the data support the conclusions?

Reviewer #1: Yes

Reviewer #2: Yes

2. Has the statistical analysis been performed appropriately and rigorously? 

Reviewer #1: Yes

Reviewer #2: Yes

3. Have the authors made all data underlying the findings in their manuscript fully available?

Reviewer #1: Yes

Reviewer #2: Yes

4. Is the manuscript presented in an intelligible fashion and written in standard English?

Reviewer #1: Yes

Reviewer #2: Yes

5. Review Comments to the Author

Reviewer #1: Thank you for inviting me to review this paper which describes a survey of water and sand exposures among visitors to one freshwater beach in Canada. It’s well written, the authors make it very clear that this is a pilot study, and thus findings and conclusions are naturally limited and cautious. However, the novelty and relevance of the research question(s) for international readers really needs to be addressed because at the moment it is not clear what the novelty of this work is, other than it is conducted in Canada. I provide more specific recommendations below.

1. I recommend a slight change to the title, as the work is better described as a prospective cohort pilot study.

2. The abstract should mention what kinds of illnesses were considered (gastrointestinal, respiratory, skin, ear and eye). The aims of the study should also be stated.

3. The methods section should include a description of sample size calculation/desired sample size and what this was based on. More information is needed about recruitment, for example, at what time of day were people recruited?

4. It is unclear how participant contact with algae was determined - I assume this was contact with seaweed at the beach, but algae could also refer to previous contact with harmful algal blooms, thus clarification is needed here.

5. How was ‘any sand contact’ defined? I’m not familiar with the study location, but at most beaches, people in contact with the water also have contact with sand as they walk across the beach to access the water. There could be misclassification here in terms of bathers reporting not having contact with sand, when they in fact did.

6. Table 6 needs reformatting as I couldn’t not see all the data. Figure 1 is not a histogram, it’s a bar chart.

7. I recommend a restructure of the discussion, which should start by repeating study objectives and main findings (in accordance with STROBE reporting). The detailed discussion of the merits of monetary incentives is not sufficiently interesting to start a discussion section, and this could be edited down to briefly state that studies with incentives for each participant report lower attrition rates, especially among under-represented groups (e.g. systematic reviews and meta-analyses by Jia et al 2021; Abdelazeem et al. 2022)

8. The discussion does not offer any explanation of the observation that incidence of some illnesses was slightly higher in the unexposed group compared to the exposed group (see water users and respiratory illness/skin infection, as well as eye infection in people not exposed to water and sand). Insufficient sample size, potentially misclassification bias or some unmeasured confounder/risk factor.

9. Antimicrobial resistant bacteria and fungi are mentioned in relation to sand quality and risks to health, but not the health threat the presence of these in water pose to bathers. There are several papers that could be usefully cited to evidence the reasons these microorganisms are concerning for recreational exposures.

10. The grammar is excellent. As PLOS ONE doesn't edit, I noted a few minor grammatical errors on the following lines:

- 29 - conducted a pilot of a prospective cohort study

- 124 - use a colon after the word 'beachgoers'

- 148 - remove the word 'to'

-Table 1 - acronyms should be included in the legend (applicable to other tables)

-line 263 - 'and there' should be 'and they'

Reviewer #2: Title: Recreational water exposures and illness outcomes at a freshwater beach in Toronto, Canada: a prospective pilot cohort study

Manuscript Number: PONE-D-22-34376

I appreciate the authors for their outstanding team work on such an interesting topic. Their work is novel. I have some suggestion for the authors as mentioned below.

Title: authors would entitle as “Incidence of recreational water activities and illness outcomes at a freshwater beach in Toronto, Canada: a prospective pilot cohort study”

Abstract: The authors would not describe the sentence “Water contact activities were lowest in June (39.1%), increasing in prevalence in July (57.4%) and August (70.8%).” in the abstract section because these findings were not part of their main study objectives. The abstract should include only pertinent findings.

Introduction:

Authors would detail the recreational water activities. What are those activities; how many types; and what are the commonest?

Line- 58: better to say “illnesses”

Authors confined the evidence searches only to Canada and U.S. What is the magnitude and impact of the phenomenon in the other continents?

“The study objectives were to determine characteristics of beachgoers, common water and sand exposures, the incidence of RWI, and the feasibility and recommendations for a larger, national cohort study” It is not clear why the authors included “…recommendations for a larger, national cohort study” to their objective. Recommendation must follow existing result.

Methods and materials

How did the authors identified base line health status was identified? How did authors ruled out previous exposure to bacterial or viral agents? Did authors use a lab-based investigation or used only participants’ verbal report?

What if the participant exposed to viral or bacterial sources (other than RW activities) amid the follow-up period? How did you controlled?

Is your tool validated?

Line-181: “low prevalence”. Is it incidence or prevalence in the eye of cohort study?

In the data analysis section, authors conducted cross-tabulations to compare water and sand exposures by age group, gender identity, highest education level in the household, and ethno-racial identity. However, they didn’t mention any statistical methods, such as chi-square tests or t-est to show differences in each category they used to compare.

Results:

Significant number of participants’ age of was below 16; however, authors mentioned as “To be eligible for participation, at least one household member aged 16 years or older needed to be present to conduct the survey in English” Can you explain this issue, please?

Discussion:

Authors reported that “parents and caregivers were often reluctant to take time to participate and complete surveys on the beach given their childcare duties, contributing to the responses of ‘not wanting to be bothered’ and ‘not having time” but they didn’t mentioned the average time each interview took.

Authors would limit their discussion to their research objectives. Most of the discussion, particularly the first and the second paragraphs, are not based on the objectives.

6. PLOS authors have the option to publish the peer review history of their article (what does this mean?). If published, this will include your full peer review and any attached files.

Reviewer #1: No

Reviewer #2: **Yes: **Wubishet Gezimu

---

## [Author Response · Author response to Decision Letter 0]

17 Feb 2023

Editor Comments:

Clarify the methodology part. Who was your study population? Source population? And what is your study unit? How did you select one (Woodbine beach) from the popular 11? (is popularity is a criteria? I did not think so). Please justify? What is the others beach?

We thank the editor for these comments. We have added and clarified these additional details as requested in the methods section (see revised study design and participant eligibility sections). The details on the other 10 beaches we believe are not relevant to this study, as we already pre-determined this beach as the ideal location for this study for the reasons specified in this methods section (i.e., large beachgoer population and most popular beach in the city ideal to assess recruitment feasibility). 

While collecting data, what do think about more than 1 participants present in household?

We have explained in the participant eligibility and enrollment section of the methods how we completed separate surveys for each individual beachgoer, but surveyed households together for feasibility and logistical reasons. This is also the approach that has been used in prior, similar studies we cited. 

Clearly describe the sample size calculation with rationale

Given that this was a pilot study aimed at determining feasibility of the methods for larger study, we did not have a specific sample size. Instead, we aimed to recruit as many beachgoers as possible each recruitment day to support power analysis calculations and planning for a larger study. This has been clarified in the participant eligibility and enrollment section of the methods.

Another of my concern is that about confidentiality (while emailing or calling the participants for the issue of get information)

As noted in the study design section of the methods, this study received research ethics board review and approval from Toronto Metropolitan University. This included comprehensive measures to ensure confidentiality of participants and their data. All participant identifying information was kept on secure servers, was available only to those in the research team, and was deleted after data collection was completed and the draw prizes administered. 

Did you found a participant with illness? What did you do for them?

As noted in Table 7, we found that a small percentage of participants that completed the follow-up survey reported one or more acute illnesses. The purpose of this study was not to provide treatments or interventions for such individuals. However, we did include information about recreational water quality in the city and where to find more information for interested participants on the last page of the survey, providing a link to the local public health authority website on beach water quality. 

Reviewer #1: 

Thank you for inviting me to review this paper which describes a survey of water and sand exposures among visitors to one freshwater beach in Canada. It’s well written, the authors make it very clear that this is a pilot study, and thus findings and conclusions are naturally limited and cautious. However, the novelty and relevance of the research question(s) for international readers really needs to be addressed because at the moment it is not clear what the novelty of this work is, other than it is conducted in Canada. I provide more specific recommendations below.

We thank the reviewer for these comments and recommendations. We have provided more information on the potential global relevance of this study in the revised discussion. 

1. I recommend a slight change to the title, as the work is better described as a prospective cohort pilot study.

We have adopted this suggestion. 

2. The abstract should mention what kinds of illnesses were considered (gastrointestinal, respiratory, skin, ear and eye). The aims of the study should also be stated.

We have added these details as suggested. 

3. The methods section should include a description of sample size calculation/desired sample size and what this was based on. More information is needed about recruitment, for example, at what time of day were people recruited?

Given that this was a pilot study aimed at determining feasibility of the methods for larger study, we did not have a specific sample size. Instead, we aimed to recruit as many beachgoers as possible each recruitment day to support power analysis calculations and planning for a larger study. We have added this information, along with more details on the recruitment, in the participant eligibility and enrollment section of the methods.

4. It is unclear how participant contact with algae was determined - I assume this was contact with seaweed at the beach, but algae could also refer to previous contact with harmful algal blooms, thus clarification is needed here.

We have clarified that we meant contact with seaweed, not harmful algal blooms (which are not a concern at the beach examined in this study). 

5. How was ‘any sand contact’ defined? I’m not familiar with the study location, but at most beaches, people in contact with the water also have contact with sand as they walk across the beach to access the water. There could be misclassification here in terms of bathers reporting not having contact with sand, when they in fact did.

We have clarified how this is reported in the tables. We specifically defined “sand contact” in this study as either digging in the sand and/or burying oneself in the sand. This is how the question appeared in the survey and was aggregated for analysis and reporting purposes. 

6. Table 6 needs reformatting as I couldn’t not see all the data. Figure 1 is not a histogram, it’s a bar chart.

We apologize for this, as it was our understanding that wide tables like this should be inserted as is into the manuscript file and it would display properly in the PDF form for reviewers. We have reformatted it to landscape to ensure it is visible. We appreciate that Fig 1 looks like a bar chart, but it is actually a histogram, as the beach water E. coli geometric mean was a numeric variable, with number of values equal to the number of recruitment days. 

7. I recommend a restructure of the discussion, which should start by repeating study objectives and main findings (in accordance with STROBE reporting). The detailed discussion of the merits of monetary incentives is not sufficiently interesting to start a discussion section, and this could be edited down to briefly state that studies with incentives for each participant report lower attrition rates, especially among under-represented groups (e.g. systematic reviews and meta-analyses by Jia et al 2021; Abdelazeem et al. 2022)

We appreciate these comments and have revised this part of the discussion accordingly. The discussion on incentives has been edited and reorganized to later in the discussion.

8. The discussion does not offer any explanation of the observation that incidence of some illnesses was slightly higher in the unexposed group compared to the exposed group (see water users and respiratory illness/skin infection, as well as eye infection in people not exposed to water and sand). Insufficient sample size, potentially misclassification bias or some unmeasured confounder/risk factor.

This a good point, and we have added a brief possible explanation for this finding as suggested in the discussion. 

9. Antimicrobial resistant bacteria and fungi are mentioned in relation to sand quality and risks to health, but not the health threat the presence of these in water pose to bathers. There are several papers that could be usefully cited to evidence the reasons these microorganisms are concerning for recreational exposures.

This is a good point, we have edited and revised this section to address both exposure pathways. 

10. The grammar is excellent. As PLOS ONE doesn't edit, I noted a few minor grammatical errors on the following lines:

- 29 - conducted a pilot of a prospective cohort study

- 124 - use a colon after the word 'beachgoers'

- 148 - remove the word 'to'

-Table 1 - acronyms should be included in the legend (applicable to other tables)

-line 263 - 'and there' should be 'and they'

We have made these edits as suggested. 

Reviewer #2: 

I appreciate the authors for their outstanding team work on such an interesting topic. Their work is novel. I have some suggestion for the authors as mentioned below.

We thank the reviewer for their comments and suggestions. 

Title: authors would entitle as “Incidence of recreational water activities and illness outcomes at a freshwater beach in Toronto, Canada: a prospective pilot cohort study”

We appreciate this suggestion but have decided to keep the original wording in the title as we feel it accurately describes the study and objectives. 

Abstract: The authors would not describe the sentence “Water contact activities were lowest in June (39.1%), increasing in prevalence in July (57.4%) and August (70.8%).” in the abstract section because these findings were not part of their main study objectives. The abstract should include only pertinent findings.

We have revised this part of the abstract as suggested.

Introduction:

Authors would detail the recreational water activities. What are those activities; how many types; and what are the commonest?

We have added more information on different types of recreational water activities, including their popularity in Canada. 

Line- 58: better to say “illnesses”

We have made this change. 

Authors confined the evidence searches only to Canada and U.S. What is the magnitude and impact of the phenomenon in the other continents?

We have focused more on Canada and the U.S. given that this study was conducted in Canada and the U.S. is the most comparable and closest country to Canada, so data from the U.S. is most relevant to the Canadian context. However, we’ve also added some additional outbreak data from the Netherlands to provide another country for comparison.

“The study objectives were to determine characteristics of beachgoers, common water and sand exposures, the incidence of RWI, and the feasibility and recommendations for a larger, national cohort study” It is not clear why the authors included “…recommendations for a larger, national cohort study” to their objective. Recommendation must follow existing result.

We have edited this sentence to remove “recommendations” as suggested. 

Methods and materials

How did the authors identified base line health status was identified? How did authors ruled out previous exposure to bacterial or viral agents? Did authors use a lab-based investigation or used only participants’ verbal report?

We collected baseline health status and all other data via beachgoer self-reported surveys. Being a pilot study, the goal of this study was to assess common exposures and outcomes and to determine feasibility of recruitment methods, not to determine causal estimates of the exposure on the outcomes. Nevertheless, as noted in the data analysis section, we excluded those who reported an illness at baseline from the summary of RWI outcomes reported in Table 7 to exclude those who might have been exposed to microbial hazards from other sources prior to their beach visit. 

What if the participant exposed to viral or bacterial sources (other than RW activities) amid the follow-up period? How did you controlled?

As noted above, this was a pilot study that did not aim to determine causal estimates of the exposure on the outcomes. It was primarily designed as a feasibility study and to assess and categorize common exposures and outcomes at the study site. Therefore, we did not conduct regression modelling to determine causal estimates, we only reported the raw incidence of each outcome for descriptive purposes. You are correct that there could be confounding factors associated with the exposure and outcome and that would need to be adjusted for in a model to determine the appropriate casual effect of the exposure, but that was beyond the scope of this pilot study. Nevertheless, we have added a brief explanation to the discussion section to indicate that confounding factors might explain any of the differences noted in Table 7. 

Is your tool validated?

As noted in the “data collection surveys” part of the methods, our questionnaire was pre-tested with 10 individual beachgoers using a cognitive interviewing technique prior to use in the current study. Additionally, the tool was based on previous validated U.S. cohort studies. Given that this is a pilot study, its use in this study provides further validation for its adoption in a larger national study in Canada. 

Line-181: “low prevalence”. Is it incidence or prevalence in the eye of cohort study?

Good point, we have revised to incidence.

In the data analysis section, authors conducted cross-tabulations to compare water and sand exposures by age group, gender identity, highest education level in the household, and ethno-racial identity. However, they didn’t mention any statistical methods, such as chi-square tests or t-est to show differences in each category they used to compare.

We decided to focus on descriptive cross-tabulations instead of null-hypothesis significance testing, as we did not have any pre-specified hypotheses to test for statistical significance. Additionally, there is a propensity for such tests and P values to be misinterpreted (e.g., https://doi.org/10.1007/s10654-016-0149-3). Instead, differences can be noted by visually examining the contingency tables, and we have aimed to highlight and comment on the key findings in the results text. 

Results:

Significant number of participants’ age of was below 16; however, authors mentioned as “To be eligible for participation, at least one household member aged 16 years or older needed to be present to conduct the survey in English” Can you explain this issue, please?

This statement means that children and youth were only permitted to participate in the study if a parent or guardian aged 16 years or older in their household was also present to provide consent for them. This has been clarified in the study design section of the methods. 

Discussion:

Authors reported that “parents and caregivers were often reluctant to take time to participate and complete surveys on the beach given their childcare duties, contributing to the responses of ‘not wanting to be bothered’ and ‘not having time” but they didn’t mentioned the average time each interview took.

We have added details on the survey completion time to the results section as suggested.

Authors would limit their discussion to their research objectives. Most of the discussion, particularly the first and the second paragraphs, are not based on the objectives.

We appreciate these comments and have revised the discussion accordingly.

---

## [Decision Letter · Decision Letter 1]

8 May 2023

PONE-D-22-34376R1Recreational water exposures and illness outcomes at a freshwater beach in Toronto, Canada: a prospective cohort pilot studyPLOS ONE

Dear Dr. Young,

Thank you for submitting your manuscript to PLOS ONE. After careful consideration, we feel that it has merit but does not fully meet PLOS ONE’s publication criteria as it currently stands. Therefore, we invite you to submit a revised version of the manuscript that addresses the points raised during the review process. Please submit your revised manuscript by Jun 22 2023 11:59PM. If you will need more time than this to complete your revisions, please reply to this message or contact the journal office at plosone@plos.org. Please include the following items when submitting your revised manuscript:A rebuttal letter that responds to each point raised by the academic editor and reviewer(s). You should upload this letter as a separate file labeled 'Response to Reviewers'.A marked-up copy of your manuscript that highlights changes made to the original version. You should upload this as a separate file labeled 'Revised Manuscript with Track Changes'.An unmarked version of your revised paper without tracked changes. You should upload this as a separate file labeled 'Manuscript'.If applicable, we recommend that you deposit your laboratory protocols in protocols.io to enhance the reproducibility of your results. Protocols.io assigns your protocol its own identifier (DOI) so that it can be cited independently in the future. For instructions see: https://journals.plos.org/plosone/s/submission-guidelines#loc-laboratory-protocols. Additionally, PLOS ONE offers an option for publishing peer-reviewed Lab Protocol articles, which describe protocols hosted on protocols.io. Read more information on sharing protocols at https://plos.org/protocols?utm_medium=editorial-email&utm_source=authorletters&utm_campaign=protocols.

We look forward to receiving your revised manuscript.

Kind regards,

Timothy J Wade, Ph.D

Academic Editor

PLOS ONE

Journal Requirements:

Additional Editor Comments:

The authors have adequately addressed all comments by the reviewers. I have a few additional comments that should be addressed in a revised version:

1) The authors conclude that a larger national cohort study is needed, but they do not address one finding of the pilot study- namely at the site they studied, the water was likely to be much too clean to see observable health relationships. Any larger study should encompass a wider range of water quality, including sites that do not have excellent water quality and sites that are impacted by sources of human sewage which are the riskiest for beachgoers. Also note that this study may be considerably expensive so the benefits should outweigh the costs

2) This study only considered limited measures of water quality, there are many advancement in water quality testing including human markers, rapid methods, and viral indicators. Any large national study should consider these as well

3) The conclusion that routine monitoring of sand quality should be considered based on this study seems premature- simply because more people are in contact with sand does not necessarily mean it should be monitored which could be considerably expensive, and may come at the trade off of other types of water quality testing. I do not know anywhere where sand is currently regularly monitored, although WHO provided a provisional value, this has not really been tested or validated broadly. Please reconsider this recommendation, at least until results of a larger study provide more conclusive evidence

Reviewers' comments:

Reviewer's Responses to Questions

**Comments to the Author**

1. If the authors have adequately addressed your comments raised in a previous round of review and you feel that this manuscript is now acceptable for publication, you may indicate that here to bypass the “Comments to the Author” section, enter your conflict of interest statement in the “Confidential to Editor” section, and submit your "Accept" recommendation.

Reviewer #2: All comments have been addressed

2. Is the manuscript technically sound, and do the data support the conclusions?

Reviewer #2: Yes

3. Has the statistical analysis been performed appropriately and rigorously? 

Reviewer #2: Yes

4. Have the authors made all data underlying the findings in their manuscript fully available?

Reviewer #2: Yes

5. Is the manuscript presented in an intelligible fashion and written in standard English?

Reviewer #2: Yes

6. Review Comments to the Author

Reviewer #2: All comments were well addressed. Authors gave point-by-point responses to my comments and queries.

The manuscript is suitable for publication in the current status.

7. PLOS authors have the option to publish the peer review history of their article (what does this mean?). If published, this will include your full peer review and any attached files.

Reviewer #2: **Yes: **Wubishet Gezimu

---

## [Author Response · Author response to Decision Letter 1]

10 May 2023

Editor Comments:

1) The authors conclude that a larger national cohort study is needed, but they do not address one finding of the pilot study- namely at the site they studied, the water was likely to be much too clean to see observable health relationships. Any larger study should encompass a wider range of water quality, including sites that do not have excellent water quality and sites that are impacted by sources of human sewage which are the riskiest for beachgoers. Also note that this study may be considerably expensive so the benefits should outweigh the costs

We agree with these points – we had a brief statement about this at the end of the limitations section, but have moved this earlier in the discussion where we discuss the water quality results, and have expanded on it accordingly. 

2) This study only considered limited measures of water quality, there are many advancement in water quality testing including human markers, rapid methods, and viral indicators. Any large national study should consider these as well

This is also a good point and we have added this to our suggestions for follow-up research in the discussion. 

3) The conclusion that routine monitoring of sand quality should be considered based on this study seems premature- simply because more people are in contact with sand does not necessarily mean it should be monitored which could be considerably expensive, and may come at the trade off of other types of water quality testing. I do not know anywhere where sand is currently regularly monitored, although WHO provided a provisional value, this has not really been tested or validated broadly. Please reconsider this recommendation, at least until results of a larger study provide more conclusive evidence

We have revised this recommendation as suggested, to instead suggest that further research is needed on possible sand contamination and its role in causing RWI. See the modified paragraph in the discussion and revised statement in the conclusion.

Note that the reference list has been slightly altered to accommodate a few additional articles cited as part of this revised discussion, and to update bibliographic details of a couple of the other references.

---

## [Editor Report · Decision Letter 2]

19 May 2023

Recreational water exposures and illness outcomes at a freshwater beach in Toronto, Canada: a prospective cohort pilot study

PONE-D-22-34376R2

Dear Dr. Young,

We’re pleased to inform you that your manuscript has been judged scientifically suitable for publication and will be formally accepted for publication once it meets all outstanding technical requirements.

Kind regards,

Timothy J Wade, Ph.D

Academic Editor

PLOS ONE
---

## [Editor Report · Acceptance letter]

23 May 2023

PONE-D-22-34376R2 

Recreational water exposures and illness outcomes at a freshwater beach in Toronto, Canada: a prospective cohort pilot study 

Dear Dr. Young:

I'm pleased to inform you that your manuscript has been deemed suitable for publication in PLOS ONE. Congratulations! Your manuscript is now with our production department. 

Kind regards, 

on behalf of

Dr. Timothy J Wade 

Academic Editor

PLOS ONE